

# Crucial role of obliquely propagating gravity waves in the quasi-biennial oscillation dynamics

Young-Ha Kim[1,a,*], Georg Sebastian Voelker[1], Gergely Bölöni[2], Günther Zängl[2], and Ulrich Achatz[1]

[1]Goethe University Frankfurt, Frankfurt am Main, Germany
[2]Deutscher Wetterdienst, Offenbach, Germany
[a]now at: Seoul National University, Seoul, South Korea
[*]Correspondence: Young-Ha Kim (kyha0830@snu.ac.kr)

**Abstract.** In climate modelling, the reality of simulated flows in the middle atmosphere is largely affected by the model's representation of gravity wave processes that are unresolved, while these processes are usually simplified to facilitate computations. The simplification commonly applied in existing climate models is to neglect wave propagation in horizontal direction and time. Here we use a model that fully represents the propagation of unresolved waves in all directions, thereby elucidating
its dynamical effect upon the climate mode in the tropical stratosphere, namely the quasi-biennial oscillation. Our simulation shows that the waves at the equatorial stratosphere, which are known to drive this climate mode, can originate far away from the equator in the troposphere. The obliquely propagating waves toward the equator are found to play a huge role in the phase progression of the quasi-biennial oscillation as well as in its penetration into the lower stratosphere. Such waves will require further attention, given that current climate models are struggling to simulate the quasi-biennial oscillation down to the lower
stratosphere to reproduce its observed impacts on the surface climate.

## 1 Introduction

Atmosphere models simulate flows on scales bounded by their resolutions, while the effects of smaller-scale unresolved processes on the simulated flows are taken into account by additional formulations, so-called parameterizations, based on our knowledge of such processes. In climate modelling, atmospheric gravity waves (GWs), an internal wave mode with horizontal
wavelengths of about 1–1000 km, are subject to parameterization, which play a pivotal role in large-scale circulations and their variability in the stratosphere and above (Fritts and Alexander, 2003; Kim et al., 2003). Their most important process in this regard is to transport momentum from the troposphere to upper layers through wave propagation, and therefore GW parameterizations primarily are to represent this process. As a simplification, existing GW parameterizations conventionally consider the wave propagation to be purely vertical and steady in time (e.g., Lindzen, 1981; Warner and McIntyre, 1999; Hines, 1997;
Scinocca, 2003), while in the real atmosphere, the propagation is oblique and transient. Effects of this usual simplification on modelled atmospheric circulations and climate variability are however not well known.

The quasi-biennial oscillation (QBO) (Ebdon and Veryard, 1961; Baldwin et al., 2001) is the prominent climate mode of the tropical stratosphere. It is characterized by persistent alternations of the flow direction between easterly and westerly, which are driven by momentum transported primarily by GWs (e.g., Dunkerton, 1997; Kawatani et al., 2010; Ern et al., 2014;





Kim and Chun, 2015). This oscillation also propagates downward to the tropopause layer, and has a broad impact in atmospheric circulations such as the stratospheric polar vortex (Holton and Tan, 1980), extratropical surface climate (Marshall and Scaife, 2009; Gray et al., 2018), and tropical convection (Gray et al., 2018; Haynes et al., 2021; Yoo and Son, 2016). The atmospheric modelling community has strived to reproduce the QBO in climate simulations and seasonal predictions (e.g., Butchart et al., 2018; Richter et al., 2020; Coy et al., 2022). Currently, many climate models are able to simulate this oscillation with reason-

able periods, using GW parameterizations tuned to supply the required momentum forcing. However, the models exhibit a common bias, i.e., a significant underestimation of the QBO-easterly magnitude in the lower stratosphere (Bushell et al., 2022; Anstey et al., 2022a). Probably related to this defect, climate models could not properly reproduce the aforementioned tropospheric impacts of the QBO (Anstey et al., 2022b; Martin et al., 2023). Moreover, the simulated QBO shows large deviations among models in its spatial structure and future evolution (Richter et al., 2020, 2022). This discrepancy as well as the common

bias in current models may reflect a lack of our knowledge in detailed dynamics of the QBO.

    Here we perform a climate simulation of the QBO using a unique GW parameterization, Multi-Scale Gravity Wave Model (MS-GWaM, see Sect. 2.2), newly developed to represent the 3-dimensional and transient wave propagation (referred to as 3d-TR experiment). The simulation result is compared to a control experiment in which the conventional simplification of GW parameterization (purely vertical and steady propagation) is applied (1d-ST experiment). Our results, for the first time, present

the role of obliquely propagating GWs on the QBO dynamics that has been veiled by the usual simplification of existing GW parameterizations. These waves are found to provide momentum forcing required especially for the descent and amplification of the easterly QBO phase in the lower stratosphere where the aforementioned common bias of climate models exists.

## 2   Methods

### 2.1   Experimental design

All the experiments use a common setup, except the use of simplifications in the GW parameterization. The ICOsahedral Non-hydrostatic model (ICON) (Zängl et al., 2015), the German operational modelling system for numerical weather prediction and climate modelling, is used (version 2.6.4-nwp5). For the study, we replace its original non-orographic GW parameterization (Scinocca, 2003; Orr et al., 2010) with the newly developed 3-dimensional transient parameterization MS-GWaM (Sect. 2.2). In addition, a 4th-order vertical damping of divergence is implemented (added to the horizontal damping of divergence established

in ICON) instead of using the existing 2nd-order background vertical diffusion in ICON. Suppressing the latter is found to be beneficial in simulating the QBO with less artificial vertical damping in the stratosphere.

    The experiments are performed with climatological-mean annual-cycle forcing (e.g., ozone, sea-surface temperature) for recent decades, for the purpose of simulating mean characteristics of the QBO over its cycles (rather than capturing its variations among the cycles). Each simulation is for 20 years after about 2 years of a spin-up period. We use a horizontal grid spacing

of $\sim$160 km (20,480 horizontal grid cells) with 180 vertical layers up to the 120 km altitude. The sponge-layer damping is applied from 85 km above. The vertical grid spacing is 400 m constantly from mid-troposphere to mid-stratosphere (36 km), and slowly increases above ($\sim$1.2 km at the sponge-layer bottom).





The experiments of the study differ only in the GW parameterization: one fully representing the 3-dimensional, transient
wave propagation (3d-TR experiment), and another applying the conventional simplifications that have been used in climate
models, i.e., representing only the vertical propagation with the steady-state assumption (1d-ST experiment). Additionally, an
experiment with the 1-dimensional but transient parameterization (1d-TR experiment), as an intermediate-level simplification,
is also performed and briefly explained in Appendix B. The different treatments in the wave-propagation modelling in these
experiments are described in Sect. 2.2 and Sect. 2.3. It should be noted that at the wave generation in the troposphere, the
parameterized wave spectra are virtually the same between the experiments in the climatological mean (refer to Fig. A1), and
therefore any differences in the simulated QBO between the experiments are due entirely to the wave-propagation modelling.

## 2.2 Gravity-wave parameterization: 3-dimensional

A GW parameterization that models 3-dimensional transient wave dynamics, MS-GWaM, has recently been developed using
a Lagrangian ray-tracing approach and implemented into ICON (Bölöni et al., 2021; Kim et al., 2021; Voelker et al., 2023).
Its detailed theoretical basis can be found in Achatz (2022); Achatz et al. (2023). Below we briefly describe its governing
equations for modelling the wave propagation.

For GWs at a position $\boldsymbol{x}$ and time $t$, their frequencies $\omega$ and wavenumbers $\boldsymbol{k}$ obey the following dispersion relation

$$\omega = \boldsymbol{U} \cdot \boldsymbol{k} + \sqrt{\frac{N^2 |\boldsymbol{k}_{\mathrm{h}}|^2 + f^2 (k_z^2 + \Gamma^2)}{|\boldsymbol{k}|^2 + \Gamma^2}} \equiv \Omega(\boldsymbol{k}, \boldsymbol{x}, t) \tag{1}$$

with $\boldsymbol{k}_{\mathrm{h}}$ and $k_z$ being respectively the horizontal and vertical components of $\boldsymbol{k}$, where $f$ is the Coriolis parameter, and all the
flow variables, i.e., horizontal wind $\boldsymbol{U}$, Brunt–Väisälä frequency $N$ and pseudo-incompressible scale-height parameter $\Gamma^{-1}$,
are functions of $(\boldsymbol{x}, t)$. The equations for modelling wave propagation consist of the ray equations

$$(\dot{\boldsymbol{x}}, \dot{\boldsymbol{k}}) = (\nabla_{\boldsymbol{k}} \Omega, -\nabla_{\boldsymbol{x}} \Omega) \tag{2}$$

to predict the position and wavenumber changes following GW rays, and the equation for wave-action density $\mathcal{N}(\boldsymbol{k}, \boldsymbol{x}, t)$ in
the 6-dimensional phase space spanned by $\boldsymbol{x}$ and $\boldsymbol{k}$

$$\frac{\mathrm{D}\mathcal{N}}{\mathrm{D}t} = \left( \frac{\partial}{\partial t} + \dot{\boldsymbol{x}} \cdot \nabla_{\boldsymbol{x}} + \dot{\boldsymbol{k}} \cdot \nabla_{\boldsymbol{k}} \right) \mathcal{N} = \mathcal{S}. \tag{3}$$

The wave-action density is conserved in that space, up to the source or sink $\mathcal{S}$ arising from wave generation or dissipation.

In the parameterization, the wave-action field is discretized spatially and spectrally into finite volumes in the phase space
(so-called ray volumes), and Eqs. (2) and (3) are solved for each ray volume in a Lagrangian manner. From the predicted $\mathcal{N}$
field, all the fields that are required to calculate the wave effects on the model flow, such as momentum fluxes and forcing
presented in Figs. 3 and 6, can be derived. Details of the discretization and the calculation of wave effects as well as the wave
dissipation modelling can be found in Voelker et al. (2023). In the 3d-TR experiment, we use about 40,000 ray volumes per
model-grid column and time at most, for accurate modelling.

The tropical source of waves taken into account by the parameterization is cumulus convection which is also parameterized,
independently, by ICON's subgrid cumulus scheme. The formulation of convectively generated GW spectra and its implemen-





tation into our parameterization for the source of $\mathcal{N}$ generally follow Song and Chun (2005) and Kim et al. (2021), respectively.
A difference exists in the present implementation compared to that work, as documented in Appendix A.

### 2.3  Gravity-wave parameterization: 1-dimensional

The 1-dimensional transient parameterization (Bölöni et al., 2021; Kim et al., 2021), which neglects the horizontal propagation, uses the same equations and methods as those described in Sect. 2.2, except applying $\dot{\boldsymbol{x}}_\mathrm{h} = \dot{\boldsymbol{k}}_\mathrm{h} = 0$ to the equations (where $\boldsymbol{x}_\mathrm{h}$ denotes the horizontal position of a wave). We use the same number of ray volumes in 1d-TR experiment as in 3d-TR
experiment ($\sim$40,000 per model-grid column and time at most).

From the 1-dimensional equations, the steady-state approximation is further applied in the 1d-ST experiment, neglecting local time derivatives. Denoting the vertical group velocity $c_{gz} = \dot{z}$ (with $z$ being the vertical coordinate) and using a general property of rays in phase-space ($\nabla_{\boldsymbol{x}} \cdot \dot{\boldsymbol{x}} + \nabla_{\boldsymbol{k}} \cdot \dot{\boldsymbol{k}} = 0$), Eq. (3) reduces to a diagnostic equation

$$\frac{\partial}{\partial z}\{c_{gz}\mathcal{N}\} = \{\mathcal{S}\} \tag{4}$$

by integration over $k_z$ for a given $\boldsymbol{k}_\mathrm{h}$ at $z$, where $\{\cdot\}$ denotes the integral. The widely used equation form in conventional GW parameterizations, which is also used in our 1d-ST experiment, is obtained accordingly as

$$\frac{\partial\{\boldsymbol{\mathcal{F}}_\mathrm{p}\}}{\partial z} = \boldsymbol{S}_\mathrm{p} \tag{5}$$

by defining pseudo-momentum $\boldsymbol{\mathcal{P}} = \boldsymbol{k}_\mathrm{h}\mathcal{N}$ with its vertical flux $\boldsymbol{\mathcal{F}}_\mathrm{p} = c_{gz}\boldsymbol{\mathcal{P}}$, where $\boldsymbol{S}_\mathrm{p} = \{\boldsymbol{k}_\mathrm{h}\mathcal{S}\}$ is the source or sink of pseudo-momentum. Therefore, the parameterization with the 1-dimensional steady-state approximation reduces to modelling
wave source and sink at every horizontal position and time.

### 3  Results

#### 3.1  Modelled structure of the quasi-biennial oscillation

The vertical and latitudinal profiles of the QBO winds in the simulations are shown in Figs. 1 and 2, respectively, along with those in the reanalysis ERA-Interim (ERA). In the vertical profiles (Fig. 1), a couple of differences are found between the
two experiments: (i) periods of the oscillation are much longer in 1d-ST (3–4 years) than those in 3d-TR (2 years), and (ii) the downward propagation of easterly phases is less pronounced in 1d-ST, exhibiting slower descents and weaker easterly amplitudes between $\sim$27 km and 19 km. Westerly phases, on the other hand, show comparable speeds of descent between the experiments until the descents halt, while afterwards they are prolonged at $\sim$21 km in 1d-ST until the easterly phases above penetrate down to this altitude. The contrast in the simulated QBO periods therefore results from the different speeds
of easterly-phase progression. Compared to ERA, the periods and peak amplitudes of the QBO are overall well reproduced in 3d-TR, while the easterly jets tend to be a bit weaker at 21–24 km.

The latitudinal profiles of the winds exhibit another notable difference between the experiments. As found above, the easterly QBO phases penetrate well down to the altitudes below 27 km in 3d-TR. Accordingly, the wind structure with alternating

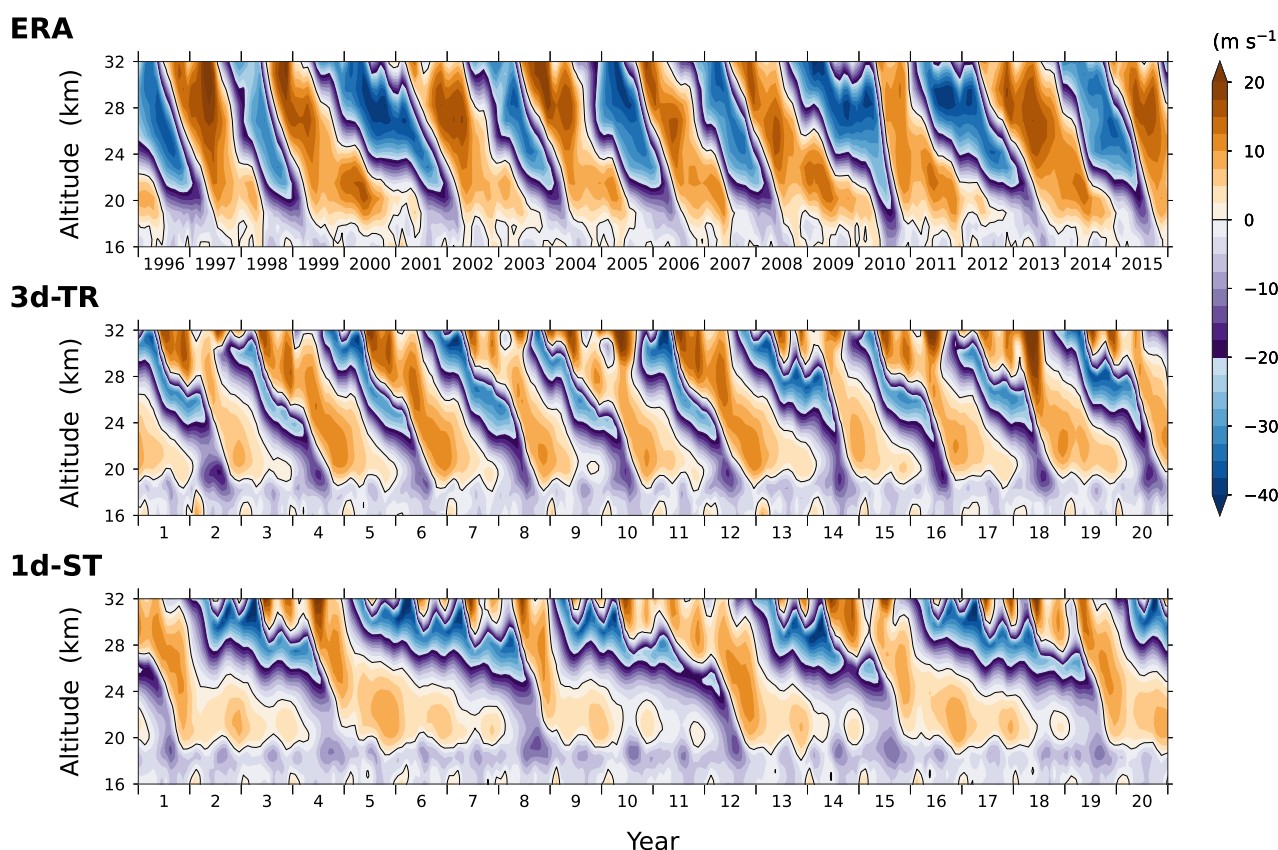

**Figure 1.** Time series of vertical profiles of the tropical stratospheric zonal winds averaged over 5° N–5° S in the two experiments, respectively using the 3-dimensional transient gravity-wave parameterization (3d-TR) and using the parameterization simplified by the conventional (1-dimensional steady-state) approximation (1d-ST), along with those in the reanalysis ERA-Interim (ERA) for 20 years. The winds have been averaged monthly and zonally. The simulations are designed to represent the climate of recent decades around the year 2000, and accordingly the time series in ERA are plotted for the period centered on the decade of the 2000s.




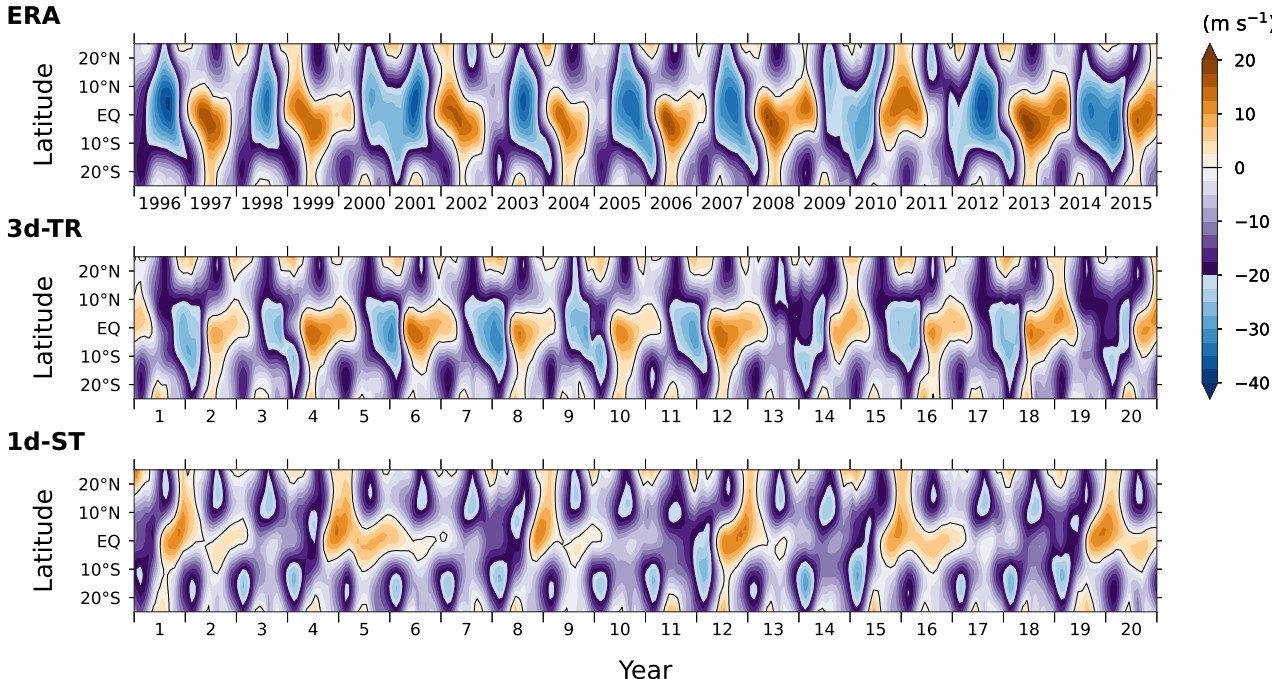

**Figure 2.** As in Fig. 1 but for the time series of latitudinal profiles at 24 km altitude.

directions around the equator is reproduced at 24 km in agreement with that in ERA (Fig. 2). In 1d-ST in contrast, as the
equatorial QBO easterlies are too weak, peak easterlies appear 10°–20° off the equator in the summer hemisphere (i.e., the
southern hemisphere at the beginning of each year and the northern hemisphere six months later). Furthermore, their magni-
tudes are overestimated by $\sim$10 m s$^{-1}$, compared to those in ERA and 3d-TR at the same locations. The result in Figs. 1 and
2 demonstrates that the simplified representation of GW propagation can lead to different latitudinal and vertical structures of
the tropical stratospheric flow in climate simulations.

**3.2    Oblique propagation of gravity waves**

For an interpretation of the above findings, we first examine GW propagation in 3d-TR. Since the major differences in the
QBO characteristics between the two experiments are associated with the easterly-phase descents (Fig. 1), we focus on easterly
momentum carried by GWs, which is responsible for these descents. Figure 3 presents horizontal fields of upward fluxes of
easterly momentum at the altitudes of 14 and 24 km (filled and open contours, respectively), due to GWs generated by tropical
convection occurring in a 1 h time window on a day in June, as an example. These target waves have been decomposed based
on their initial horizontal wavelengths (i.e., wavelengths at generation) in Fig. 3. The flux fields are integrated in time, so that
they would be approximately conserved up to wave dissipation between the two altitudes if the wave propagation were purely





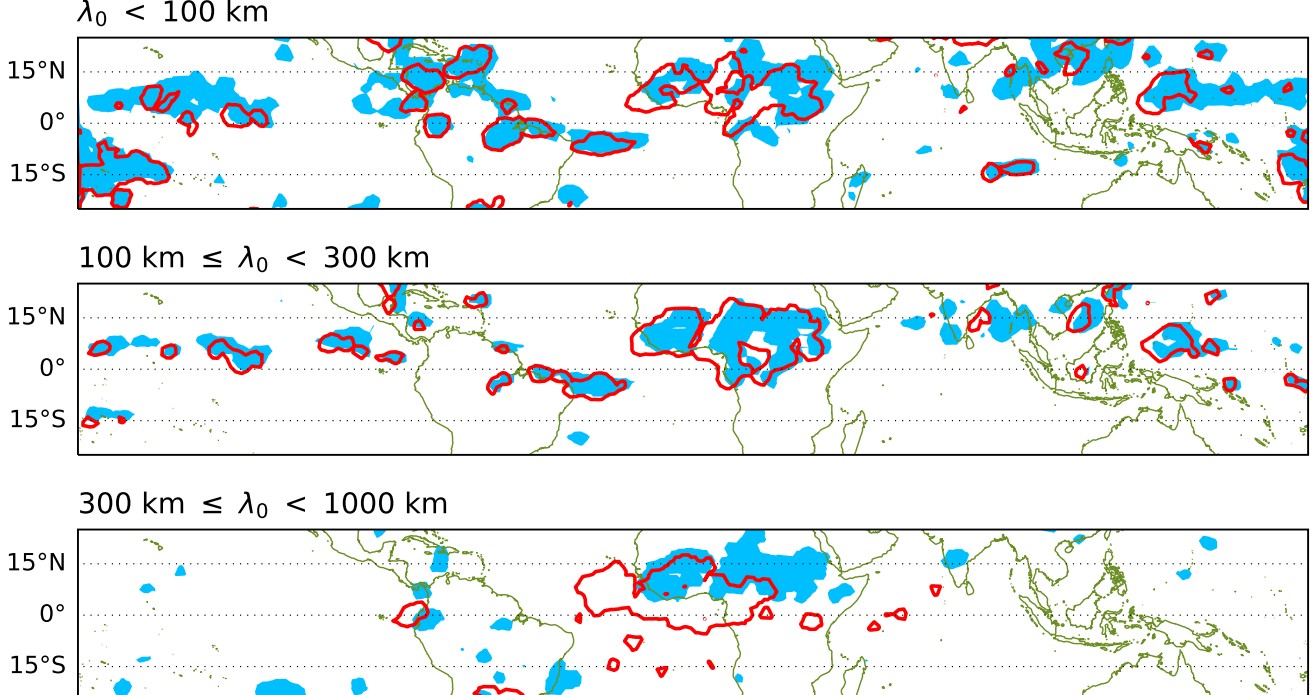

**Figure 3.** Horizontal fields of time-integrated upward fluxes of easterly momentum (contoured at 0.5 mPa h) due to gravity waves parameterized in 3d-TR experiment, at two altitudes, 14 km (blue, filled) and 24 km (red, open), for comparison. Only the waves that are generated during a certain time window (for 1 h on a day in June) are taken into account to trace the given waves' displacement, and they are decomposed based on the horizontal wavelengths at their generation ($\lambda_0$): $\lambda_0 < 100$ km, $100$ km $\leq \lambda_0 < 300$ km, and $300$ km $\leq \lambda_0 < 1000$ km (from top to bottom). The fluxes are integrated over a period long enough (4 d) to cover the entire wave propagation up to the 24 km altitude.

vertical. Therefore, changes in the flux distribution with altitude indicate oblique propagation of the waves, along possibly with the wave dissipation effect. In particular, the waves with horizontal wavelengths larger than 300 km observed over Africa at the 14 km altitude are found to propagate southwestward, by up to about 15° until they reach the 24 km altitude (Fig. 3). In contrast, waves with wavelengths smaller than 300 km travel much less in horizontal directions ($\lesssim 5°$) which are mostly westward.

Such equatorward slanted propagation over considerable distances as seen for the case in Fig. 3 occurs preferentially at a particular phase of the QBO but persistently in every QBO cycle throughout the 20-year simulation period. Figure 4 shows zonally averaged upward fluxes of easterly momentum due to GWs (shading) along with zonal winds (contours), composited for each QBO phase during the 20 years. As the QBO in 3d-TR has regular 2-year periods (Fig. 1), we define its phases simply by eight of consecutive 3-month periods for each QBO cycle, such that the first phase (Phase 1) corresponds to the period with the maximum easterly wind being located at about 31 km ($\sim 10$ hPa) altitude (refer to the zonal-wind fields in Fig. 4). Phase 1 corresponds to February–April of every other year.




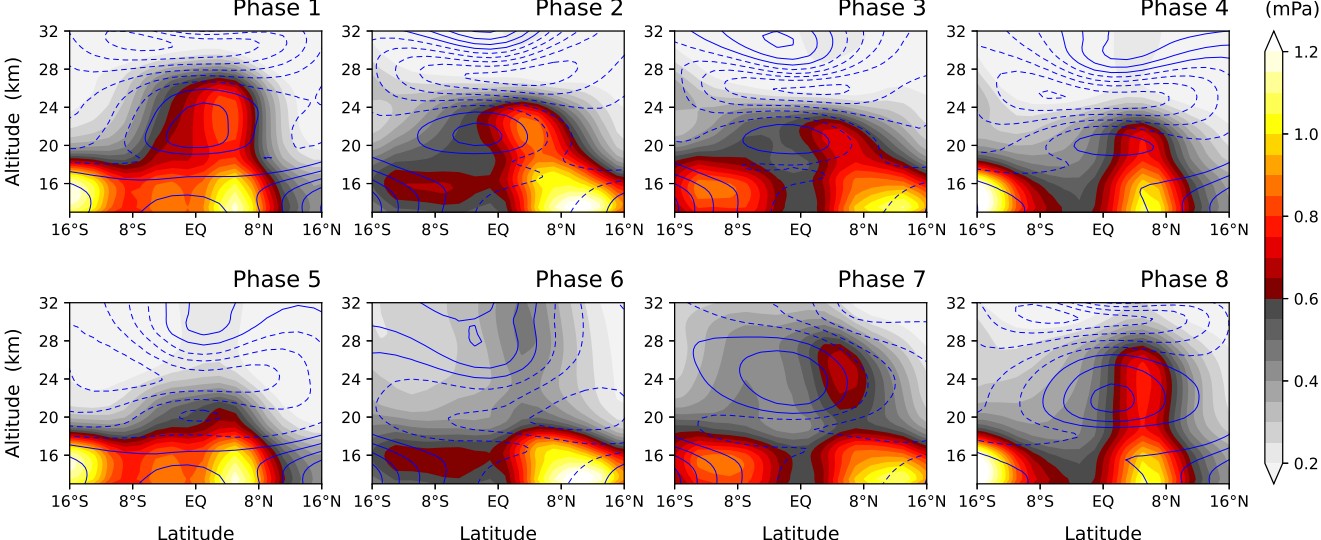

**Figure 4.** Composite mean of zonally averaged easterly-momentum fluxes due to gravity waves (shading) along with zonal winds (blue contours with dashed lines for easterly winds and solid lines for zero and westerly winds, at the intervals of $5\,\mathrm{m\,s^{-1}}$) in 3d-TR experiment, for each QBO phase (Phase 1 to Phase 8). In this experiment, the QBO phases are defined by consecutive 3-month periods, with the first one (Phase 1) being the period when the maximum easterly wind is located at $\sim$31 km altitude ($\sim$10 hPa). Phase 1 corresponds to February–April of every other year (note that the periods of the QBO in 3d-TR experiment are 2 years regularly).

In general, the easterly-momentum fluxes in the tropical upper troposphere ($\sim$15 km) are broadly distributed with latitude, and their maxima are often located off the equator (Fig. 4), following the seasonal dependence of convection. In the stratosphere, the momentum fluxes tend to decrease with altitude in westward sheared layers throughout the QBO cycle, due to wave dissipation. Oblique propagation of waves is manifested during Phase 2–3 by a slanted structure of the fluxes. In particular, the equatorward propagation (as also observed for the example given in Fig. 3) can be identified, originating from around 10° N

in the upper troposphere. Such equatorward propagation is however not evident in the other phases of the QBO.

The propagation path of waves is controlled by their ambient wind structure, which the QBO modulates, as well as by their own characteristics (Lighthill, 1978). Our simulation shows that waves carrying easterly momentum (i.e., waves with westward intrinsic phase velocities) tend to propagate obliquely when the ambient flow is weakly easterly in the upper troposphere to lower stratosphere, as during Phase 2–3 presented in Fig. 4. This condition is satisfied when the QBO easterly is maximal in

the middle stratosphere (Fig. 4). On the other hand, the waves propagate more vertically in westerly ambient flows (e.g., during Phase 8–1) or tend to dissipate in vertically sheared flows when the easterly QBO phase has descended to the lower stratosphere (Phase 5–7). These behaviours are qualitatively consistent with the theory that GWs are modulated to propagate more vertically than horizontally where the ambient flow velocity backs away from the phase velocities of the waves, i.e., where the intrinsic





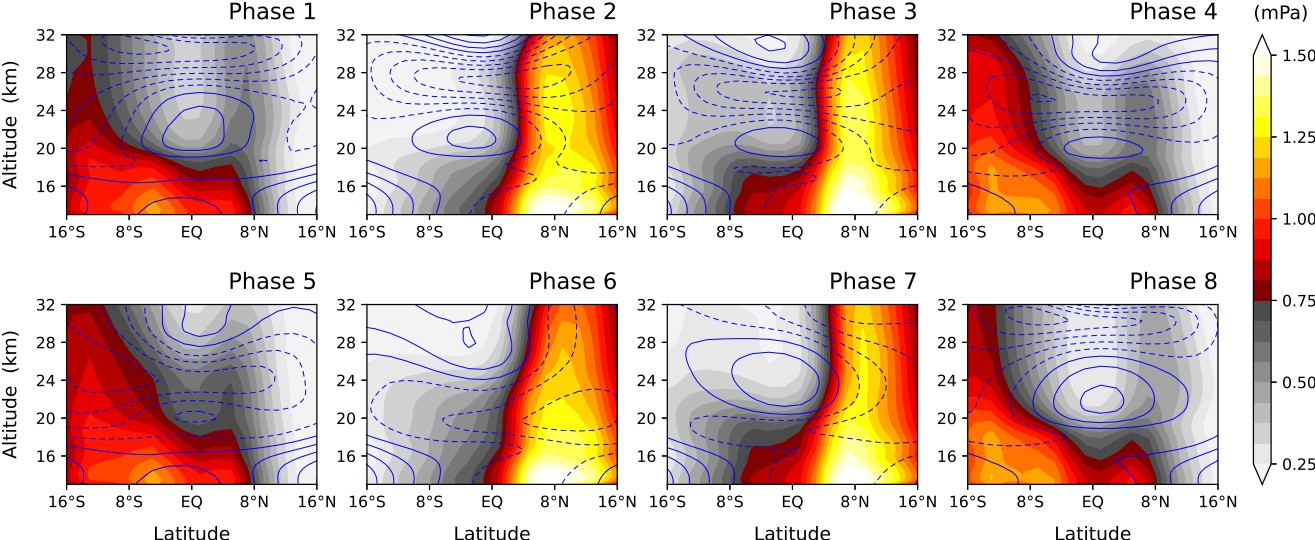

**Figure 5.** As in Fig. 4 but for westerly-momentum fluxes.

phase velocities are increased (e.g., Lighthill, 1978). It is found from further investigations that GWs travelling long distances

toward the equator in the lower stratosphere generally have horizontal wavelengths larger than about 300 km (not shown).

The westerly-momentum fluxes composited for the eight QBO phases are shown in Fig. 5 for completeness. Here, the fluxes peaking off the equator do not exhibit the slanted structure. This may be attributed to the easterly ambient flows in the stratosphere off the equator, which modulate the waves that carry westerly momentum (waves with eastward intrinsic phase velocities) to propagate more vertically.

### 3.3 Effect of oblique wave propagation on the QBO

The persistent occurrences of the equatorward propagation during Phase 2–3 suggest that it may robustly play a role in the QBO dynamics. Here we investigate the GW forcing of the QBO around Phase 2–3 in 3d-TR and 1d-ST. Phase 2 in 3d-TR corresponds to May–July of every other year where the easterly maximum wind is located at around 28 km (see Sect. 3.2 and Fig. 4). Consistently, Phase 2 in 1d-ST is defined as the 3-month period with the easterly maximum being located around this

altitude in each QBO cycle. For comparison to 3d-TR, only those cycles where Phase 2 corresponds to May–July (three cycles out of five in 1d-ST) are considered in the following composite analysis. Figure 6 shows the easterly-momentum fluxes and zonal-wind forcing due to GWs (shading and green contour, respectively) during Phase 2 (center panels), along with those over the consecutive 3-month periods before and after Phase 2 (left and right panels, respectively), in 3d-TR and 1d-ST (upper and lower, respectively).

In 1d-ST, by construction, the wave propagation is purely vertical, and the momentum fluxes only decrease with altitude where waves dissipate. The GW forcing of zonal winds typically occurs where the vertical gradient of the flux is large. In





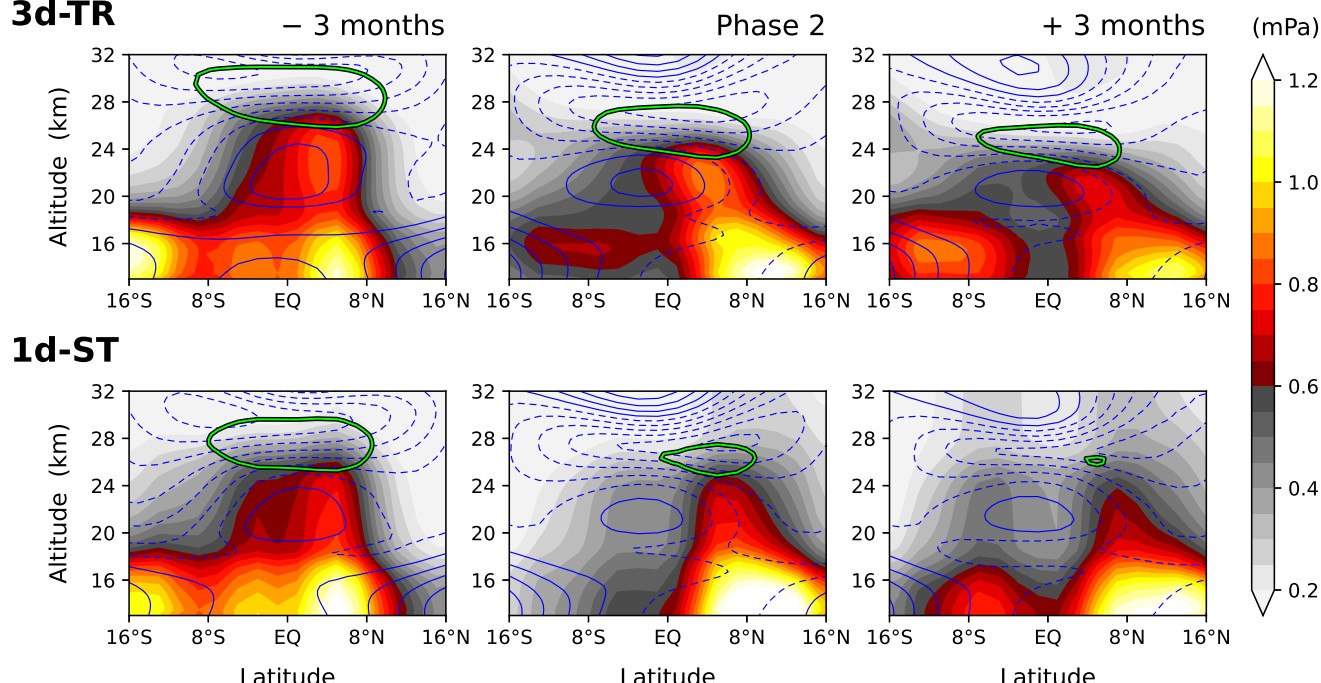

**Figure 6.** As in Fig. 4 but for Phase 2 in 3d-TR and 1d-ST experiments (upper center and lower center, respectively), with zonal-mean zonal momentum forcing due to gravity waves (green contour, at $-0.2\,\mathrm{m\,s^{-1}\,d^{-1}}$) being superimposed. In 1d-ST experiment, Phase 2 is defined as the 3-month period where the easterly maximum is located at about 28 km in each QBO cycle, but only the cycles in which Phase 2 corresponds to May–July are composited here, so that the features in the same season and phase are compared between the two experiments. The consecutive 3-month periods before and after Phase 2 are shown in the left and right panels, respectively, in each experiment (which, in 3d-TR experiment, are Phases 1 and 3, as in Fig. 4). The numbers of the composited QBO cycles are 10 and 3 in 3d-TR and 1d-ST experiments, respectively.

3d-TR, three months before Phase 2, the overall distribution of the momentum fluxes and forcing is similar to that in 1d-ST (Fig. 6, upper left and lower left). However, during Phase 2 when the equatorward wave propagation is manifested, the momentum fluxes at about 24 km altitude exhibit their maximum around the equator, and they strongly dissipate higher up

due to the large shear associated with the equatorial QBO jet (Fig. 6, upper center). This induces substantially large easterly-momentum forcing below the easterly-maximum altitude, thereby leading to the descent of the easterly maximum afterwards (see Fig. 6, upper right). This behaviour is in strong contrast with 1d-ST where the momentum forcing occurs off the equator with a weaker magnitude during Phase 2 and therefore the easterly descent is much slower. This result demonstrates that the descent of the easterly QBO phase is largely affected by the wave propagation path, explaining the differences in the speed of

descent and vertical penetration between 3d-TR and 1d-ST shown in Figs. 1 and 2.





**1d-ST-tuned**

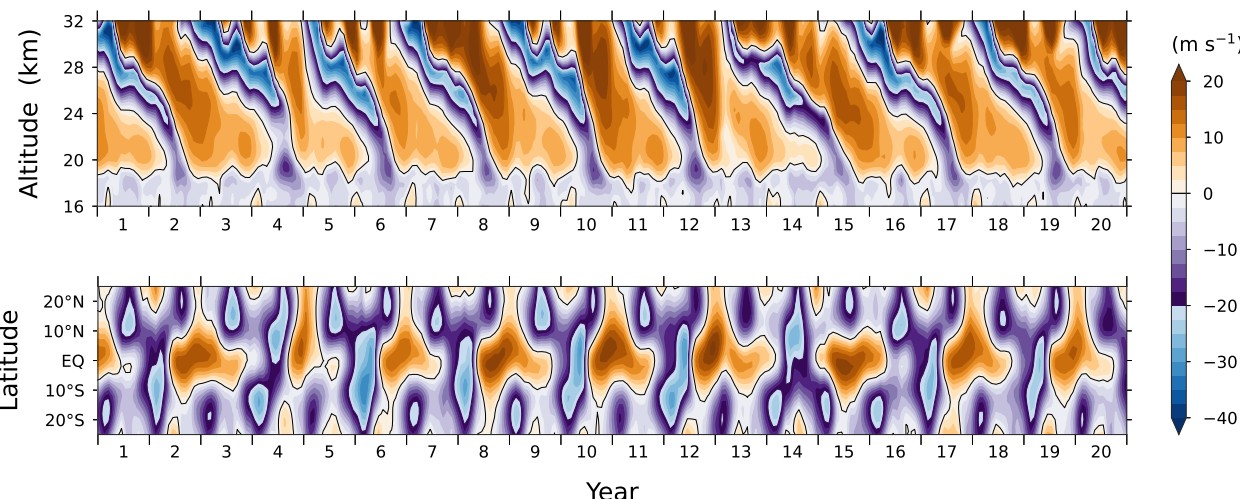

**Figure 7.** Time series of vertical profiles (upper) and 24 km latitudinal profiles (lower) of the tropical stratospheric zonal winds in the experiment using 1d-ST parameterization but tuned by raising the launching fluxes of gravity waves by 50 % in order to obtain realistic periods of the oscillation (2–3 years).

## 3.4 Comparison to the 1-dimensional parameterization being tuned

Our results show that, via oblique propagation, waves that originate off the equator provide the equatorial stratospheric flow with momentum which significantly accelerates the QBO. In climate modelling with conventional 1-dimensional GW pa-rameterizations, a practical and general approach to accelerate the QBO has been to empirically enhance the magnitude of
momentum flux of waves at their launch locations over the equator so that the required momentum can be supplemented above in the stratosphere. While a reasonable time scale of the oscillation could be acquired by this approach, the spatial structures of the modelled flows should be examined in comparison to those resulting from the realistic oblique wave propagation.

Following the approach, we repeat the 1d-ST simulation but with GW fluxes increased by 50 % (empirically determined) at launch locations, and compare its result (Fig. 7) to 3d-TR. The periods of the QBO in this experiment are modelled to be
2–3 years as intended, due to fast descents of easterly phases (Fig. 7). However, the easterly phases descend less in depth, having shorter phase durations than those in 3d-TR, while westerlies are too strong above ∼23 km (cf. Fig. 1). In the summer hemisphere in the lower stratosphere, the excessive easterly bias found in 1d-ST still remains with similar magnitudes (cf. Fig. 2). The discrepancy of these results from 3d-TR reflects the fact that the oblique equatorward propagation of waves in 3d-TR occurs and accelerates the QBO preferentially during the easterly-descending phase of the oscillation, whereas the simple
tuning in the 1-dimensional parameterization accelerates or amplifies the entire phases and also over-accelerates flows off the equator (e.g., in the summer hemisphere) to model reasonable periods of the equatorial oscillation. Given this physical reason,



it is convincing that such a discrepancy would remain even if another climate model was used for the current study, although some quantitative details would change. In addition, to mimic the effects of realistic wave propagation somehow using a 1-dimensional parameterization, its tuning will need to be designed in a sophisticated way, based on the understanding of actual
processes of GWs.

## 4 Discussion

Although not presented in this study, we document that ICON with its original GW parameterization (Scinocca, 2003; Orr et al., 2010), which uses a spatio-temporally uniform, prescribed wave spectrum and a different wave-dissipation scheme from that used here, simulated the QBO with generally weak amplitudes when the same experimental setup as described in Sect. 2.1 was
used.

The obliquely propagating waves that significantly affect the QBO in 3d-TR have horizontal wavelengths of 300–1000 km with variable vertical wavelengths down to ~1 km. Waves on these scales are subject to parameterization, as they are not fully resolved by current climate models due to the limitation in horizontal and vertical resolutions as well as to the difficulty in properly generating the wave source (multi-scale convection, such as mesoscale convective systems). In our simulation, the
waves on those scales account for only about 10 % of the parameterized GW spectrum in the tropics (see Fig. A1 for the spectrum). Given their large effects on the QBO (Figs. 1 and 2) even with the relatively small contribution to the spectrum, quantitative observational investigations of them will be required to better understand and model the QBO. It may still be improbable to explicitly capture 3-dimensional GW propagation using current measurement techniques. Nonetheless, a recent observational campaign (Haase et al., 2018) produced statistics showing that a substantial portion of tropical GWs detected in
the lowermost stratosphere (~20 km) had their sources at far horizontal distances (~10°) in the troposphere (Corcos et al., 2021), which supports our simulation result of oblique propagation.

It is especially under the descending QBO-easterly phase in the lower stratosphere, where the effect of obliquely propagating waves is large in our simulation (Fig. 6), but this effect could be even larger depending on the quantitative details of the waves. The oblique wave propagation process is therefore a strong hint for the aforementioned common model bias of the lower
stratospheric QBO easterlies which needs to be corrected to reproduce the observed downward impact of the QBO on the surface climate (Anstey et al., 2022b). Finally, it should be highlighted that the QBO projection on a changing climate, which was not robustly simulated among models and/or GW parameterizations (Richter et al., 2022; Schirber et al., 2015), may be more reliable using a 3-dimensional GW parameterization because the wave propagation features vary depending on flow structures under the changing climate.

*Code and data availability.* The ICON Software is freely available to the scientific community for noncommercial research purposes under a licence of DWD and MPI-M [please contact icon@dwd.de]. The MS-GWaM code and its module for the implementation in ICON have been developed at Goethe University Frankfurt, and are available from Prof. Ulrich Achatz [achatz@iau.uni-frankfurt.de] on reasonable request.





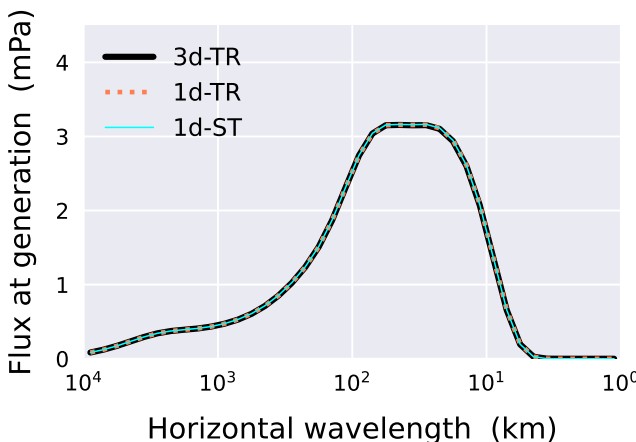

**Figure A1.** Horizontal-wavelength spectrum of vertical fluxes of absolute horizontal momentum due to gravity waves launched at 15° N–15° S over the 20-year simulation period in 3d-TR (black solid line), 1d-TR (pink dotted line), and 1d-ST experiments (sky-blue solid line).

The simulation datasets generated and analysed during the current study are available from the corresponding author. The ERA-Interim dataset is publicly available [https://doi.org/10.24381/cds.f2f5241d].

## Appendix A: Parameterized wave spectrum at generation

As documented in Sect. 2.2, the formulation and implementation of convectively generated GW spectra follow Song and Chun (2005) and Kim et al. (2021). In the present implementation, however, a notable difference exists from that work. While there for the horizontal and temporal scales of convective latent heating ($\delta_\mathrm{h}$, $\delta_t$), which are preset parameters used in the source formulation, a single scale set has been taken (5 km and 20 min for horizontal and temporal scales, respectively), here a distinctly larger-scale set (100 km, 12 h) is used in addition, in order to take the multi-scale nature of tropical convection into account. The latter scale is chosen as a representative scale of convective heating distribution in mesoscale convective systems that are unresolved by climate models (e.g., Tao and Moncrieff, 2009), and it is found to be important to generate waves that have wavelengths larger than ∼300 km in our simulations (refer to Trinh et al., 2016). The calculated spectrum at wave generation using those two scale sets, averaged over the tropics for the whole simulation period, is presented in Fig. A1.

## Appendix B: 1d-TR experiment

While the conventional simplification applied in 1d-ST consists of the two approximations (1-dimensional and steady-state propagation), the impact of oblique wave propagation examined in Sect. 3 should also be confirmed by exclusively applying the 1-dimensional simplification but with transient GW parameterization. An additional experiment performed with this simplification (1d-TR, Fig. B1) shows qualitatively similar results to 1d-ST, also exhibiting too long periods of the oscillation



**1d-TR**

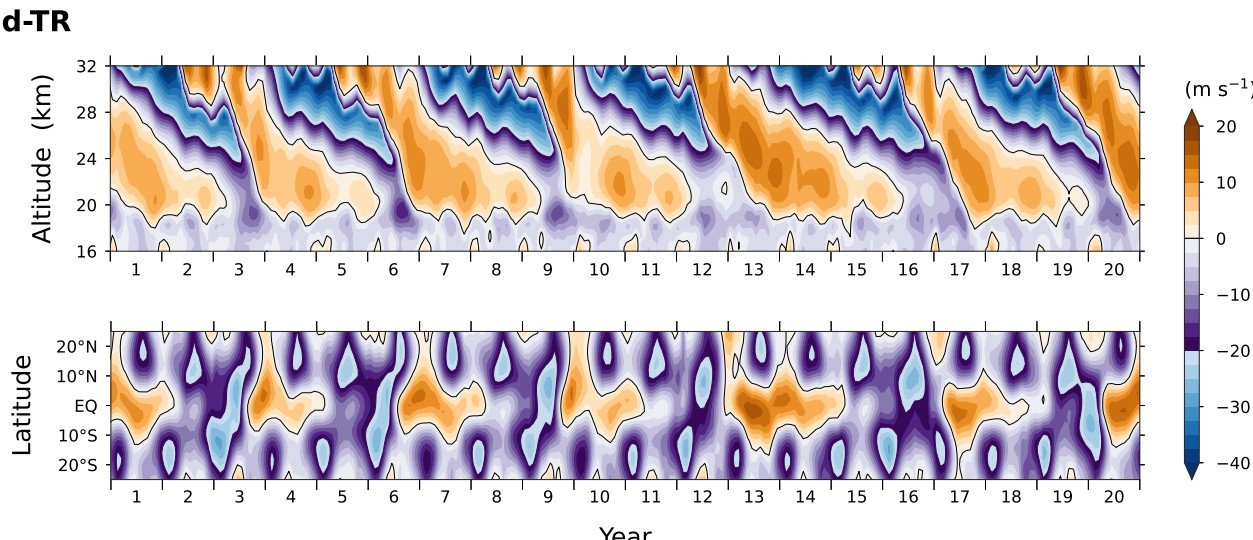

**Figure B1.** Time series of vertical profiles (upper) and 24 km latitudinal profiles (lower) of the tropical stratospheric zonal winds in the experiment using the transient gravity-wave parameterization simplified by the 1-dimensional approximation (1d-TR).

(3–4 years) with slow downward penetration of the easterly phase, and the excessive easterly bias at 10°–20° latitudes of the summer hemisphere.

*Author contributions.* All authors contributed to the development of the model MS-GWaM. YHK designed and carried out experiments and analyses. All authors extensively discussed the results and implications. YHK prepared the manuscript with contributions from all co-authors.

*Competing interests.* The authors declare that they have no conflict of interest.

*Acknowledgements.* UA thanks the German Research Foundation (DFG) for partial support through the research unit "Multiscale Dynamics of Gravity Waves" (MS-GWaves, grants AC 71/8-2, AC 71/9-2, and AC 71/12-2) and CRC 301 "TPChange" (Project-ID 428312742, Projects B06 "Impact of small-scale dynamics on UTLS transport and mixing" and B07 "Impact of cirrus clouds on tropopause structure"). YHK and UA thank the German Federal Ministry of Education and Research (BMBF) for partial support through the programme "Role of the Middle Atmosphere in Climate" (ROMIC II: QUBICC) and through grant 01LG1905B. UA and GSV thank the German Research Foundation (DFG)

for partial support through CRC 181 "Energy transfers in Atmosphere and Ocean" (Project Number 274762653, Projects W01 "Gravity-wave parameterization for the atmosphere" and S02 "Improved parameterizations and numerics in climate models"). UA is furthermore grateful





for support by Eric and Wendy Schmidt through the Schmidt Futures VESRI "DataWave" project. This work used resources of the Deutsches Klimarechenzentrum (DKRZ) granted by its Scientific Steering Committee (WLA) under project ID bb1097.



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
