# Peer review of "Crucial role of obliquely propagating gravity waves in the quasi-biennial oscillation dynamics"

_EGUsphere, 2023_

## Author Comment (AC1)

**Author Responses to the Referee Comments**

We deeply appreciate the three referees for their helpful comments on the manuscript. Our responses to each comment, along with the revised line numbers in the manuscript, are provided below in blue font.

**1 RC1**

The manuscript titled "Crucial role of obliquely propagating gravity waves in the quasi-biennial oscillation dynamics" (https://doi.org/10.5194/egusphere-2023-2663) offers significant insights into the dynamics of the Quasi-Biennial Oscillation (QBO) in the stratospheric wind field. The introduction of horizontally propagating gravity waves in climate models, as discussed in the paper, highlights their vital role in more realistically simulating the QBO. This contributes to a better simulation of the QBO and aids in understanding the stratosphere-troposphere coupling and the role of gravity waves in climate change.

However, several aspects may require further elaboration or modification:

 $\gg~$  We appreciate Referee 1 for providing constructive comments which will improve our manuscript.

1. Order of Figures: The first figures mentioned in the text are Figures 3 and 6 (Line 84). Describing the figures in the order they appear in the manuscript is recommended for better logical flow and reader comprehension.

 $\gg~$  The text in the manuscript has been revised [L85] to refer to figures in the order in which they appear.

2. Data Introduction: The ERA dataset used in the study is mentioned (Line 108). It would be beneficial to provide a more detailed introduction to this dataset, including its source, characteristics, and reasons for its selection.

 $\gg$  We have added Sect. 2.4, which provides a description of the ERA dataset as suggested by the referee, to the revised manuscript.

3. Analysis in Figure 4: The discussion on the easterly momentum of Phase 2 and Phase 3 of QBO involves obliquely propagating gravity waves. Please clarify the definition of oblique propagation and consider whether the reduction in gravity wave flux between 8-16N with altitude is related to the absorption by the background easterlies (critical level filtering). Additionally, could the structure of the background wind and its differential absorption of gravity waves contribute to the inclined structure of the wave flux? This aspect might be further investigated by providing more information on zonal mean zonal wind forcing in Figure 6.

 $\gg$  Thank you for the careful review of the analysis. Our definition of the oblique propagation follows that stated in Sect. 3.2 [L144]: propagation leading to a change in the horizontal distribution of the fluxes with altitude (up to wave dissipation). Regarding the comment on the flux reduction at 8–16° N and the relation between the structures of wave flux and background wind, we have added an explanation in the revised manuscript [L161–163].

4. Gravity Wave Flux Phenomenon: In the context of the QBO's second and seventh phases, the manuscript describes a phenomenon where gravity wave flux weakens and strengthens with altitude. Does this suggest that the gravity waves at 24 km might not necessarily propagate from lower altitudes and that there could be independent wave sources above 20 km?

 $\gg$  No, it does not. All the analyzed fluxes in our paper are due to parameterized gravity waves which have their source solely within the troposphere (Sect. 2.2). The increase in fluxes with altitude, shown in Phase 7 with a localized maximum aloft, could potentially be attributed to waves converging in the meridional direction. However, confirming this would require a complicated analysis using additional model diagnostics. Therefore, we do not provide insight into this aspect in Phase 7 but rather focus on the slanted flux structure during other phases. A similar localized maximum is indeed also observed in Phase 2, albeit less distinctly.

5. English Writing: Lastly, further improvement in English writing is suggested to enhance the overall readability and coherence of the manuscript.

 $\gg~$  We have checked the text and tried our best to improve it throughout the revised manuscript.

**2 RC2**

This manuscript reports on the implementation of a 3D gravity wave (gw) parameterization that accounts for obliquely propagating GW on the ICON model. They compare it with ICON simulations using a 1D gw parameterization. They specifically report at how ICON simulations the QBO using the two different parameterization. Their results clearly indicate improved simulations with the 3D gw parameterization. They also show how the gw momentum variations reacts to the QBO winds. My main concern is the interpetation of the results on how the 3D gw parameterization impacts the QBO winds. The interpretations assert that all of the changes in the QBO winds are solely and directly from gw momentum improvements. Zonal winds throughout the atmosphere are driven by momentum from a myriad of sources. For the QBO winds in particular, we cannot dismiss momentum from Kelvin and Mixed-Rossby Gravity Waves in the model. In line 184-185 for example, it is argued that the descent of the easterly QBO phase is largely affected by the wave propagation path. To argue this, one would need to first show quantitatively that the momentum of the winds in the region is predominantly from gravity waves and secondarily from other sources. I agree that the gw parameterization induced changes to the QBO but whether it is directly or indirectly is still not clear from these results. Hence, I suggest revising these interpretations. Other than that, I recommend accepting this paper pending minor revisions.

>> We appreciate Referee 2 for the insightful suggestion. We have added the discussion regarding this comment in the revised manuscript [L199–207]. In our simulation, the resolved wave forcing of the zonal-mean wind is an order of magnitude smaller than the parameterized GW drag over the equator during the easterly-descending phase (refer to Fig. R1 showing the resolved wave forcing around Phase 2; cf. green contour in Fig. 6). This implies that even though the resolved waves (Rossby–gravity waves and planetary-scale gravity waves, during this phase) could be altered by the different representations of GWs, their impact on the easterly-phase descent must be relatively minor. Meanwhile, Kelvin waves in the opposite phase induce momentum forcing with a magnitude comparable to the GW forcing, but our result does not exhibit a distinct difference in the descent of the westerly phases between 3d-TR and 1d-ST, on average (Fig. 1).

Minor comment: In section 3.2, kindly expand on how you calculated the fluxes. Show the relevant equations. Also, how did you perform the filtering of the waves?

 $\gg$  The flux formulation is given in the revised manuscript [L138; L174]. The filtering into the easterly-momentum and westerly-momentum fluxes is based on the sign of intrinsic zonal phase velocity, which is now stated along with the flux formulation.

Figure R1. Eliassen–Palm flux divergence due to model-resolved waves for Phase 2 (center) and for the consecutive 3-month periods before and after Phase 2 (left and right, respectively) in 3d-TR experiment.

**3 RC3**

This study delves into the importance of the oblique propagation of off-equator gravity waves in the formation of the Quasi-Biennial Oscillation (QBO) by comparing the results of a 3D gravity wave model and a 1D gravity wave model. The insights provided are valuable not only to understanding QBO dynamics but also hold relevance for future QBO modeling. The clarity of analysis and figures in this paper is commendable, and I recommend its acceptance with only minor revisions.

 $\gg$  We would like to express our appreciation to Referee 3 for the evaluation and helpful suggestions. We have incorporated all the suggested changes into the revised manuscript.

**Comments:**

- 1. Equation 1: Please clarify the meaning of  $\Omega$ .
  - $\gg$  It is now clarified in the revised manuscript [L75].

2. Line 109: When introducing ERA-Interim, consider providing details about the dataset, including its time span, temporal resolution, and a relevant citation.

- >> We have added Sect. 2.4, which provides a detailed description of the dataset as suggested by the referee, to the revised manuscript. The time span of the data and its rationale are provided in the caption of Fig. 1, as we believe this is the optimal location, preventing any potential misunderstanding of the figure.
- 3. Line 110: Instead of 'years', consider specifying the timeframe in months for greater precision.
  - $\gg$  It is revised as suggested [L119].
- 4. Line 142: When referring to 'phase 1', include a citation for QBO phases.
  - $\gg$  As the eight *nominal* phases (Phase 1 to Phase 8) follow our own definition introduced in the manuscript, there does not exist a paper to cite that used the same method in the literature. In the revised manuscript, we clarify that the numbering of phases is nominal [L153].

**Author Response to the Community Comment 1**

If the annual nodal signal (and it's harmonics) drives the Semi-Annual Oscillation (SAO) which occurs directly above the QBO in altitude, by symmetry what drives the QBO? Is it obliquely propagating gravity waves generated by the lunar nodal cycle? Likely. The numbers predict it – the lunar nodal Draconic cycle interfering with the annual cycle will generate a frequency of  $365.2 / 27.2122 \mod 13 = 0.422 \ cycles/year => 2.37 \ years, which matches the QBO cycle. These two nodal cycles, for sun and moon respectively, are the only cycles that can drive wavenumber=0 behaviors such as SAO and QBO. Any other cycles, such as synodic/tropical, have the wrong group symmetry and so would generate wavenumbers > 0 due to longitude specificity. Whereas the draconic nodal lunar orbit is invariant of longitude, leading to exclusively wavenumber=0 forcing. Perhaps this gap in understanding should finally be addressed because of the foundational importance it holds to the geophysics.$

≫ It has long and well been understood in the literature that the QBO is driven by various types of (zonally asymmetric) atmospheric waves in the tropics through the wave-mean-flow interaction (Baldwin et al., 2001). Tropical deep convection persistently generates a broad spectrum of gravity waves, including those with wavelengths of several hundred kilometers. Our paper suggests that the waves propagating obliquely are generated by the convective source. The paper also demonstrates that the propagation path of these waves is largely influenced by the background wind state which is modulated by the QBO and seasonal variability. Considering the prominence of variability attributed to the QBO and the seasons in atmospheric flows, modulation by the lunar nodal cycle may be relatively minor, although this aspect has not been examined previously. In addition, climate models have simulated the QBO with varying periods depending very sensitively on the amplitude of tropical waves. This indicates that even if the lunar nodal cycle would be involved in the QBO dynamics, it role might be rather limited. Including the lunar forcing somehow into the model is beyond the scope of the study.

**Reference:**

Baldwin, M. P., Gray, L. J., Dunkerton, T. J., Hamilton, K., Haynes, P. H., Randel, W. J., Holton, J. R., Alexander, M. J., Hirota, I., Horinouchi, T., Jones, D. B. A., Kinnersley, J. S., Marquardt, C., Sato, K., and Takahashi, M.: The quasi-biennial oscillation, Rev. Geophys., 39, 179–229, https://doi.org/10.1029/1999RG000073, 2001.